# Rep-Adapter: Parameter-free Automatic Adaptation of Pre-trained ConvNets via Re-parameterization

## Abstract

Recent advances in visual pre-training have demonstrated the advantage of transferring pre-trained models to target tasks. However, different transfer learning protocols have distinctive advantages regarding target tasks, and are nontrivial to choose without repeated trial and error. This paper presents a parameter-free automatic model adaptation protocol for ConvNets, aiming at automatically balancing between fine-tuning and linear probing, by using adaptive learning rate for each convolution filters on target tasks. First, we propose Rep-Adapter, an adapter module with re-parameterization scheme, which can achieve soft balancing between the pre-trained and fine-tuned filters, and can be equivalently converted to a single weight layer, without introducing additional parameters to the inference phase. We show by theoretical analysis that Rep-Adapter can simulate a ConvNet layer with each filter fine-tuning at different learning rate. We present a simple adapter tuning protocol with Rep-Adapter to achieve automatic adaptation of pre-trained models without additional search cost. Extensive experiments on various datasets with ResNet and CLIP demonstrate the superiority of our Rep-Adapter on semi-supervised, few-shot and full dataset transfer learning scenarios.

## 1 Introduction

Recent advances in both computer vision and neural language processing field have demonstrated the advantage of transferring large-scale pre-trained models to downstream tasks. To leverage the information of the pre-trained models, there emerges different transfer learning protocols such as fine-tuning directly on the new task (Girshick et al., 2014), feature-based methods (Turian et al., 2010) that only update the parameters of the task-specific head, and adapter tuning (Houlsby et al., 2019) that introduces new learnable modules to adapt some layers in the original model.

The most suitable choice of the protocols depends on many factors, including target dataset size, label fraction, *etc*. For instance, fine-tuning is usually more powerful on fully-supervised ImageNet, while linear probing are sometimes better on semi-supervised ImageNet (Zhou et al., 2022a). We also observe a similar trend on Caltech101 dataset. As in shown in Figure 1, fine-tuning outperforms linear probing for full dataset transfer on Caltech101 (dataset B in the figure), while linear probing performs better for low-shot (1000 examples) scenario. Moreover, when taking partial fine-tuning and layer-wise learning rate adjustment into consideration, the option space becomes computationally prohibitive for an extensive search. Therefore, we believe there is an urgent need of designing an automatic transfer learning protocol that can combine the advantages of these protocols without tediously searching for hyper-parameter settings.

There exists some preliminary attempts to avoid manually tuning layer-wise and filter-wise hyper-parameter settings. SpotTune (Guo et al., 2019) and AdaFilter (Guo et al., 2020b) dynamically route between frozen and learnable weights, at a cost of having more than twice the original model size during inference. AutoLR (Ro & Choi, 2021) searches for layer-wise fine-tuning learning rate via repeated trial. However, these methods introduce either policy networks or repeated training loops, increasing the training or inference cost. We thus explore a simple model adaptation method, without introducing extra training steps or extra parameters during inference.

In this paper, we introduce Rep-Adapter, a novel adapter module that achieves soft balancing between the pre-trained and fine-tuned filters. Each adapter module has two branches, a frozen branch

with the pre-trained parameters, and a learnable branch that adapts on the target tasks. During inference, we leverage the re-parameterization scheme to equivalently convert the two branches into one, avoiding the increase in model size and computational cost.

In addition, we prove that, during training, our Rep-Adapter is equivalent to a layer fine-tuned by a fine-grained transfer learning protocol with different learning rate for each filter. This eliminates the need of tedious hyper-parameter search for learning rate configurations like previous methods. By optimizing a network stacked by Rep-Adapter, the pre-trained model can be transferred with adaptive learning rate for each filter. As such, fine-tuning and linear probing can be achieved by learning the learning rate to a certain value or *zero*.

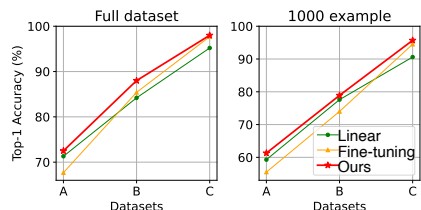

Figure 1: Comparison of fine-tuning, linear probing and our proposed Rep-Adapter. Dataset A: DTD for texture classification; B: Caltech101 for object identification; C: EuroSAT for satellite image classification. Models are pre-trained on ImageNet with a self-supervised method, PIRL.

We conduct extensive experiments to evaluate and analysis our method. We use VISSL (Goyal et al., 2021) library to evaluate the effectiveness of our approach on full dataset and low-shot (1000 examples) transfer learning scenarios. We adapt models from several different pre-training protocols (fully-supervised pre-training and three state-of-the-art self-supervised pre-training methods on ImageNet, CLIP pre-training with text supervision), to various downstream datasets, and on full-shot, low-shot (1000 examples) or few-shot (16-shot) scenarios. We also perform experiments on semi-supervised ImageNet, by adapting models pre-trained by 3 different self-supervised learning methods to ImageNet with 1% and 10% label fraction. Moreover, we explore the transferability to VOC object detection and Cityscapes, COCO instance segmentation tasks. Our Rep-Adapter tuning protocol consistently surpasses the baselines across tasks and scenarios, which clearly evidences the effectiveness and generalization ability of our method.

Overall, our contributions can be summarized as follows:

- We present a novel adapter module that has two advantages: 1) the module keeps the pre-trained model intact, avoiding the issue of catastrophic forgetting, preserving all the general knowledge learnt during pre-training; 2) in contrast to previous adapter-based tuning approach, our module can be merged into a single layer during inference to achieve a parameter-free tuning.
- We theoretically prove that, in Rep-Adapter tuning, each module achieves an automatic learning rate adjustment to avoid manual hyper-parameter search.
- Our Rep-Adapter tuning protocol surpasses the state-of-the-art transfer learning protocols by a significant margin in extensive experiments. Rep-Adapter tuning consistently outperforms fine-tuning and linear probing protocols, on different tasks and scenarios (Figure 1).

## 2 RELATED WORK

**Transfer Learning via Model Adaptation.** Model adaptation methods modify a pre-trained model to achieve higher performance on a target task. Figure 2 illustrates popular model adaptation methods in computer vision field. *Feature-based* methods (Figure 2 (a)), including linear classification, directly learn task specific head on *frozen* features (Turian et al., 2010). *Fine-tuning* (Girshick et al., 2014) is the most generally used transfer learning protocol (Figure 2 (b)). Some works regularize the distance between the fine-tuned model and the pretrained model on the network parameters (*e.g.*, $L^2$-SP (Li et al., 2018)) or the output features (*e.g.*, DELTA (Li et al., 2019)) to improve the performance, but the weight of the regularization comes a hand-crafted hyper-parameter. Some works (Furlanello et al., 2018; Huang et al., 2017) use network ensemble to boost the performance by combining the frozen and learnable weights, at a cost of using $M \times$ model size ($M$ is the number of ensemble models) during inference. *Adapter tuning* (Yin et al., 2023; Hu et al., 2023; Zhang et al., 2023; Houlsby et al., 2019; Yuan et al., 2020) (Figure 2 (c)) adds light-weight modules on the pre-trained network to adapt the model without changing its original parameters. Previous works on convolution adapter modules (Rebuffi et al., 2017; 2018; Rosenfeld & Tsotsos, 2018) perform incremental learning by adding small point-wise convolution adapter modules, at a cost of about 11% increase in model size for each tasks. Our Rep-Adapter is also related to ensemble methods. However, an ensemble of models can not be directly converted to a single model due to the non-linearity in the models and thus much larger than us in terms of parameters and computation complexity. Different from previous methods, Rep-Adapter does not increase inference model size, which accelerates the deployment and removes the size constraint of the adapter modules.

Table 1: Comparison of different model adaptation protocols. (*orig. weight*: accessible to original weight; *auto.*: automatic protocol; *no ext. train*: no extra training steps; *no ext. infer.*: no extra inference cost)

| Method | orig. weight | auto. | no ext. train | no ext. infer. |
|---|---|---|---|---|
| Fine-tuning | ✗ | ✗ | ✓ | ✓ |
| Feature-based | ✓ | ✗ | ✓ | ✓ |
| Adapter | ✓ | ✗ | ✓ | ✗ |
| SpotTune | ✓ | ✓ | ✗ | ✗ |
| AutoLR | ✗ | ✓ | ✗ | ✓ |
| Rep-Adapter | ✓ | ✓ | ✓ | ✓ |

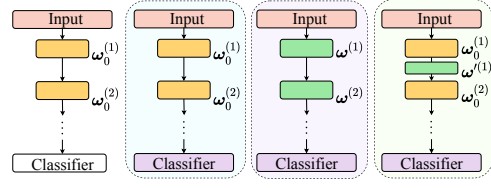

Figure 2: **Illustration of different model adaptation protocols.** Feature-based protocols train classifier on top of the frozen feature. Fine-tuning train all the parameters. Adapter tuning adds learnable adapters. (Orange: frozen, green: learnable)

Some methods including LoRA (Hu et al., 2022) and Progressive Networks (Rusu et al., 2016) also adopt the idea of using both frozen branches and learnable branches in the network. The main differences between Rep-Adapter and these works are summarized in Tab. 2. **First**, different from LoRA and Progressive Networks, the added branch in Rep-Adapter is non-linear during training and linear during inference. This is in line with vision re-parameterization works which benefit from the unique property of BN layer. **Second**, unlike these works, we add two scaling factor layers to the two branches and prove that our module achieves an automatic learning rate adjustment to avoid manual hyper-parameter search. **Third**, different from LoRA, Rep-Adapter does not aim at parameter-efficient tuning of large pretrained models, but aims at the transfer learning performance, in line with works such as L2-SP (Li et al., 2018), DELTA (Li et al., 2019), *etc*.

**AutoML on transfer learning.** There exists some preliminary attempts on automating transfer learning. Dynamic tuning methods (Guo et al., 2019; 2020b; Yang et al., 2021) introduce more than $2\times$ parameters during inference, with additional policy networks or gating modules to achieve dynamic routing among frozen and fine-tuning weights. AutoLR (Ro & Choi,

Table 2: Comparison of Rep-Adapter, Progressive Network and LoRA.

| Method | train linearity | inference linearity | learnable factor |
|---|---|---|---|
| Progressive | ✗ | ✗ | ✗ |
| LoRA | ✓ | ✓ | ✗ |
| Rep-Adapter | ✗ | ✓ | ✓ |

2021) presents auto-tuning of layer-wise learning rates, at a cost of performing repeated learning rate trial during training. Different from these methods, we solve automatic model adaptation problem in a much simpler way, without introducing extra training steps, or extra model size during inference. A comprehensive comparison of different model adaptation schemes is shown in Tab. 1. We consider four aspects including: preservation of original weight, automatic method, no extra training steps and no extra inference cost. Among these methods, only ours could cover all aspects.

**Traditional Re-parameterization.** Traditional Re-parameterization (Zagoruyko & Komodakis, 2017) parameterize weight as a combination of a set of parameters for optimization of deep neural networks. Traditional Re-parameterization has also been used for AutoML. (Chen et al., 2019) uses meta kernels with re-parameterization to denote a neural architecture search space. SCC (Yang et al., 2019b) and CondConv (Yang et al., 2019a) derive a kernel as the weighted sum of multiple generated kernels, to perform a conditionally parameterized convolution. Differently, we focus on automating the optimization of learning rate during transfer learning.

**Structural Re-parameterization.** Structural Re-parameterization, proposed by (Ding et al., 2021c), is a method that parameterizes a network structure with the parameters transformed from another network structure. This method has recently been used to boost the training phase of ConvNets (Luo et al., 2023; Ding et al., 2021b; 2019; Guo et al., 2020a; Cao et al., 2020) and MLP models (Ding et al., 2021a) by introducing merge-able branches with different architectures, without changing the whole network architecture in the inference phase. In this paper, we use re-parameterization to represent structural re-parameterization. Different from these works that merge branches with different architecture, we use re-parameterization to merge frozen branch and tuning branch (with the same architecture), and simulate a layer with arbitrary learning rate. This is the first time that re-parameterization is proved to be able to represent different learning rate, and also the first time that scaling factors are introduced to re-parameterization for learning rate optimization.

## 3 METHODOLOGY

In this section, we first formulate the automatic model adaptation problem and analyze the downside of baseline methods, then introduce *Rep-Adapter*, a novel adapter module with re-parameterization technique. We prove that our adapter can simulate a fine-tuning layer with arbitrary learning rate for each filter, and further present a simple adapter tuning protocol to achieve automatic adaptation of pre-trained models without additional search cost.

### 3.1 PRELIMINARIES

**Model Adaptation.** Let $\psi$ denote a pre-trained model equipped with a random initialized task specific head. Given a target task with objective $\{\min \mathcal{L}\}$, model adaptation yields a target model $\psi^\star$ by executing certain transfer protocol $\zeta$:

$$\psi^\star = \arg\min_{\psi} \mathcal{L}(\psi, \zeta). \qquad (1)$$

In computer vision field, most of the transfer learning protocols for ConvNets use the same network architecture of the pre-trained model on the target task, where the network parameters $\omega$ inherit from the pre-trained ones $\omega_0$. As discussed in Section 1, most of these protocols fall in two categories, fine-tuning and linear probing. The essential difference between these protocols is the learning rate of each network parameters. Without loss of generality, let $\eta_m$ be a reference learning rate, and $\eta \in \mathbb{R}^{|i|}$ be the learning rate multipliers $\eta^{(i)} \in \mathbb{R}$ for all the $|i|$ filters (where $i$ indexes network filters and $|i|$ denotes total filter number). Given the hyper-parameter $\eta$, the objective of these protocols is:

$$\omega^\star = \arg\min_{\omega} \mathcal{L}(\omega, \eta). \qquad (2)$$

In this framework, the fine-tuning protocol is equivalent to setting a global $\eta$ for all parameters, while linear probing is equivalent to setting $\eta = 0$.

**Automatic Model Adaptation.** Intuitively, the optimal learning rate of each parameter is influenced by factors like data distribution, model architectures. To this end, another line of work aims to learn the proper transfer protocol $\zeta$ for the target task in an automatic fashion. Following the previous discussion, the objective of automatic model adaptation can be denoted as:

$$\min_{\omega, \eta} \mathcal{L}(\omega, \eta). \qquad (3)$$

In the following sections, we first introduce a novel parameter-free adapter module, and then present a carefully designed tuning protocol that is equivalent to a bi-level optimization (Anandalingam & Friesz, 1992) of the learning rate and network parameters.

### 3.2 PARAMETER-FREE ADAPTATION VIA REP-ADAPTER

Intuitively, a soft balancing between a frozen filter and a fine-tuning filter can represent an intermediate state between them, *i.e.*, a filter tuning at a smaller learning rate. Motivated by the intuition, we set to solve this problem by first proposing an adapter module to achieve this balance. This intuition will be further analysed in Sec. 3.3.

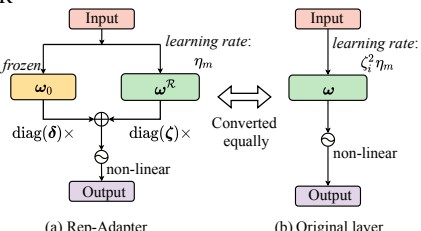

(a) Rep-Adapter.    (b) Original layer.

Figure 3: A Rep-Adapter can be equivalently converted to a single weight layer, and can also simulate a layer fine-tuning with certain filter-wise learning rate by setting the value of $\zeta$ accordingly.

**Rep-Adapter.** To make a soft balancing between a ConvNet layer with frozen filters and a ConvNet layer with tuned filters, we present the Rep-Adapter module that comprises a frozen branch and a tuning branch, both initialized with pre-trained network parameters. We explicitly introduce the scaling factors $\delta, \zeta \in \mathbb{R}^d$ for each filter of the two branches, where $d$ denotes filter number for a network layer. As shown in Figure 3 (a), Rep-Adapter can be defined as:

$$\psi^{\mathcal{R}}(\omega_0, \omega^{\mathcal{R}}) := \mathrm{diag}(\delta)\psi(\omega_0) + \mathrm{diag}(\zeta)\psi(\omega^{\mathcal{R}}), \qquad (4)$$

where $\omega_0$, $\omega^{\mathcal{R}}$ denote pre-trained network parameters and parameters of the tuning branch, and $\mathrm{diag}(\cdot)$ denotes the corresponding diagonal matrix constructed from a vector. Such soft weighting with factors $\{\delta, \zeta\}$ in the adapter would generally introduce twice model size and computation, leading to high inference cost and model size. Accordingly, we introduce a re-parameterization scheme to merge the two branches in Rep-Adapter.

**Re-parameterization.** Considering a pre-trained ConvNet, a Rep-Adapter $\psi^{\mathcal{R}}$ for the ConvNet contains a frozen branch and an adapting branch, both of which consist of a convolution (Conv) layer and a batch normalization (BN) layer. The weights and optional bias of a Conv layer can be denoted by $w$ and $b$. A BN layer performs per-channel normalization and scaling. Let $\mu$ and $\sigma$ denote running mean and variance of training samples, and let $\gamma$ and $\beta$ be the channel-wise scaling factor and bias. A Conv-BN branch can be converted to a single Conv layer as:

$$\psi(x) = (w \star x - \mu)\frac{\gamma}{\sigma} + \beta = \underbrace{(w \cdot \frac{\gamma}{\sigma})}_{w'} \star x + \underbrace{(\beta - \mu\frac{\gamma}{\sigma})}_{b'}, \qquad (5)$$

where $\star$ denotes convolution operator, $\boldsymbol{w}'$ and $\boldsymbol{b}'$ are the equivalent weights and bias after the merge.

After merging Conv-BN, we consider to merge the two Conv branches in Rep-Adapter. When the outputs of two or more Conv layers with the same configurations are added up, we can merge them into a single Conv. Let $\boldsymbol{w}_0$, $\boldsymbol{b}_0$ be the pre-trained weights and bias of a merged Conv-BN branch, and $\boldsymbol{w}^{\mathcal{R}}$, $\boldsymbol{b}^{\mathcal{R}}$ be the merged weights and bias of the tuning Conv-BN branch in Rep-Adapter. The adapter, *i.e.*, weighted sum of two branches, can be converted as:

$$\boldsymbol{\psi}^{\mathcal{R}}(\boldsymbol{x}) = \text{diag}(\boldsymbol{\delta})(\boldsymbol{w}_0 \star \boldsymbol{x} + \boldsymbol{b}_0) + \text{diag}(\boldsymbol{\zeta})(\boldsymbol{w}^{\mathcal{R}} \star \boldsymbol{x} + \boldsymbol{b}^{\mathcal{R}}). \tag{6}$$

Thus, the adapter can be merged to a single convolution layer by setting:

$$\widetilde{\boldsymbol{w}} = \text{diag}(\boldsymbol{\delta})\boldsymbol{w}_0 + \text{diag}(\boldsymbol{\zeta})\boldsymbol{w}^{\mathcal{R}}, \quad \widetilde{\boldsymbol{b}} = \text{diag}(\boldsymbol{\delta})\boldsymbol{b}_0 + \text{diag}(\boldsymbol{\zeta})\boldsymbol{b}^{\mathcal{R}}. \tag{7}$$

In this way, a Rep-Adapter $\boldsymbol{\psi}^{\mathcal{R}}$ with two branches can be converted to a single weight layer $\boldsymbol{\psi}(\widetilde{\boldsymbol{\omega}})$ during inference (Figure 3), enabling parameter-free model adaptation.

### 3.3 SIMULATION OF LAYERS FINE-TUNING AT ANY LEARNING RATE VIA REP-ADAPTER

In Sec. 3.2 we intuitively use our proposed Rep-Adapter as a balance between fine-tuning and frozen weights. Here we show by theoretical analysis that Rep-Adapter can indeed simulate a layer fine-tuning at a different learning rate.

**Proposition 3.1.** *Given a pre-trained linear network layer $\boldsymbol{\psi}(\boldsymbol{\omega}_0)$. Let $\boldsymbol{\psi}^{\mathcal{R}}(\boldsymbol{x}) = diag(\boldsymbol{\delta})\boldsymbol{\psi}(\boldsymbol{\omega}_0) + diag(\boldsymbol{\zeta})\boldsymbol{\psi}(\boldsymbol{\omega}^{\mathcal{R}})$ denotes this layer under adapter tuning scheme at learning rate $\eta_m$, and $\boldsymbol{\psi}^{\mathcal{F}}(\boldsymbol{\omega})$ denotes this layer with each of the $d$ filters fine-tuning at a certain learning rate $\boldsymbol{\eta} \in \mathbb{R}^d$. Suppose the network functions of the two schemes are identical before tuning step $t$, then there always exists scaling factor $\boldsymbol{\zeta} \in \mathbb{R}^d$ for each filter such that after one step of tuning, $\boldsymbol{\psi}^{\mathcal{F}}(\boldsymbol{x}) = \boldsymbol{\psi}^{\mathcal{R}}(\boldsymbol{x})$ still holds.*

*Proof.* Suppose the network functions of the two schemes are identical at step $t$. We have $\boldsymbol{\psi}_t^{\mathcal{F}}(\boldsymbol{x}) = \boldsymbol{\psi}_t^{\mathcal{R}}(\boldsymbol{x})$ and $\partial \mathcal{L}_t / \partial \boldsymbol{\psi}_t^{\mathcal{R}} = \partial \mathcal{L}_t / \partial \boldsymbol{\psi}_t^{\mathcal{F}}$. First, we calculate the weight change of $\boldsymbol{\omega}^{\mathcal{R}}$ in the adapter:

$$\Delta \boldsymbol{\omega}_t^{\mathcal{R}} = -\eta_m \frac{\partial \mathcal{L}_t}{\partial \boldsymbol{\omega}_t^{\mathcal{R}}} = -\eta_m \cdot \text{diag}(\boldsymbol{\zeta}) \frac{\partial \mathcal{L}_t}{\partial \boldsymbol{\psi}_t^{\mathcal{R}}}. \tag{8}$$

Let $\widetilde{\boldsymbol{\omega}}_t = \text{diag}(\boldsymbol{\delta})\boldsymbol{\omega}_0 + \text{diag}(\boldsymbol{\zeta})\boldsymbol{\omega}_t^{\mathcal{R}}$, this layer function after one-step adapter tuning is:

$$\begin{aligned} \boldsymbol{\psi}_{t+1}^{\mathcal{R}}(\boldsymbol{x}) &= \text{diag}(\boldsymbol{\delta})\boldsymbol{\omega}_0 \boldsymbol{x} + \text{diag}(\boldsymbol{\zeta})(\boldsymbol{\omega}^{\mathcal{R}} + \Delta \boldsymbol{\omega}^{\mathcal{R}})\boldsymbol{x} \\ &= (\widetilde{\boldsymbol{\omega}}_t - \eta_m \cdot \text{diag}(\zeta_1^2, \ldots, \zeta_d^2) \frac{\partial \mathcal{L}_t}{\partial \boldsymbol{\psi}_t^{\mathcal{R}}})\boldsymbol{x}. \end{aligned} \tag{9}$$

As for the fine-tuned case:

$$\boldsymbol{\psi}_{t+1}^{\mathcal{F}}(\boldsymbol{x}) = \boldsymbol{\omega}_{t+1}\boldsymbol{x} = (\boldsymbol{\omega}_t - \text{diag}(\boldsymbol{\eta})\frac{\partial \mathcal{L}_t}{\partial \boldsymbol{\psi}_t^{\mathcal{F}}})\boldsymbol{x}. \tag{10}$$

Recall that $\boldsymbol{\psi}_t^{\mathcal{F}}(\boldsymbol{x}) = \boldsymbol{\psi}_t^{\mathcal{R}}(\boldsymbol{x})$, and $\partial \mathcal{L}_t / \partial \boldsymbol{\psi}_t^{\mathcal{R}} = \partial \mathcal{L}_t / \partial \boldsymbol{\psi}_t^{\mathcal{F}}$. Thus, there exists:

$$\boldsymbol{\zeta} = \left( \sqrt{\frac{\eta_i}{\eta_m}} \right)_{i=1}^d, \quad \text{such that } \boldsymbol{\psi}_{t+1}^{\mathcal{F}}(\boldsymbol{x}) = \boldsymbol{\psi}_{t+1}^{\mathcal{R}}(\boldsymbol{x}), \tag{11}$$

which proves that after one step of tuning, $\boldsymbol{\psi}^{\mathcal{F}}(\boldsymbol{x}) = \boldsymbol{\psi}^{\mathcal{R}}(\boldsymbol{x})$ still holds. $\square$

Based on Proposition 3.1, we have two corollaries. *First*, as in Figure 3, if a Rep-Adapter and a network layer with filter-wise fine-tuning *lr* are identical before training ($\boldsymbol{\psi}_0^{\mathcal{F}}(\boldsymbol{x}) = \boldsymbol{\psi}_0^{\mathcal{R}}(\boldsymbol{x})$), by properly setting scaling factors $\boldsymbol{\zeta}$ in Rep-Adapter based on the filter-wise learning rate $\boldsymbol{\eta}$, the Rep-Adapter can represent this layer in every steps ($\boldsymbol{\psi}^{\mathcal{F}}(\boldsymbol{x}) = \boldsymbol{\psi}^{\mathcal{R}}(\boldsymbol{x})$). *Second*, when using the optimal scaling factors $\boldsymbol{\zeta}$ at each training step, the Rep-Adapter represents a layer fine-tuning with optimal filter-wise *lr*. Therefore, the optimization of *lr* can be relaxed to the optimization of $\boldsymbol{\zeta}$.[1]

---

[1]Note that BN layer during training generally uses batch statistics to simulate the data distribution. This causes non-linearity in BN and could make Proposition 3.1 not strictly true. However, we empirically find that using batch-wise statistics are better than using frozen statistics loaded from pretrained model (see Sec. 4.8 (c)). This is probably because BN with batch-wise statistics acts very similar to a linear layer. Rep-Adapter using BN with batch-wise statistics performs consistently good even when training with extremely low batchsize (*i.e.*, batchsize=2, see Sec. 4.7). To align with previous works, we use batch-wise BN statistics during training.

### 3.4 Automatic Model Adaptation via Rep-Adapter Tuning

**Rep-Adapter tuning.** We present a simple Rep-Adapter tuning scheme, by adding the Rep-Adapter to every layer of a pre-trained model, and making the factors $\boldsymbol{\delta}$ and $\boldsymbol{\zeta}$ in the adapters learnable. In Rep-Adapter tuning protocol, adapter parameters $\boldsymbol{\omega}^{\mathcal{R}}$ and factors $\{\boldsymbol{\delta}\} = \{\boldsymbol{\delta}^{(l)}\}_{l=0}^{|l|}$, $\{\boldsymbol{\zeta}\} = \{\boldsymbol{\zeta}^{(l)}\}_{l=0}^{|l|}$ of all the layers are jointly optimized. Based on Proposition 3.1, we have $\boldsymbol{\omega} = \mathrm{diag}(\boldsymbol{\delta})\boldsymbol{\omega}_0 + \mathrm{diag}(\boldsymbol{\zeta})\boldsymbol{\omega}^{\mathcal{R}}$, $\eta = \mathrm{diag}(\boldsymbol{\zeta}^2)\eta_m$. Thus, the original objective of automatic model adaptation in Equation (3) can be converted to:

$$\min_{\boldsymbol{\omega}^{\mathcal{R}},\{\boldsymbol{\delta}\},\{\boldsymbol{\zeta}\}} \mathcal{L}(\boldsymbol{\omega}_0, \boldsymbol{\omega}^{\mathcal{R}}, \{\boldsymbol{\delta}\}, \{\boldsymbol{\zeta}\}, \eta_m), \tag{12}$$

where pre-trained $\boldsymbol{\omega}_0$ and reference learning rate $\eta_m$ are task-agnostic hyper-parameters. Since $\boldsymbol{\zeta}$ participates in loss calculation, it can be simply optimized by back-propagation together with $\boldsymbol{\omega}^{\mathcal{R}}$.[2] Thus, Rep-Adapter tuning simulates a model fine-tuning with adaptive learning rate for each filter without introducing additional training loops or searching phase. In practice, the weight $\boldsymbol{\gamma}$ of BN layer of the two branches have the same shape and play a similar role to the scaling factors $\boldsymbol{\delta}$ and $\boldsymbol{\zeta}$. Thus, we directly use $\boldsymbol{\gamma}$ as the scaling factor for convenience in our implementation.

After the tuning phase, the model can be deployed efficiently by re-parameterization following the scheme in Sec. 3.2. The model size and computation complexity are exactly the same as in fine-tuning scheme, without introducing any additional parameters during inference. This scheme avoids the issue of catastrophic forgetting during training, and the tuned model can also be used for further transfer as the original weights are not modified.

## 4 Experiments

We evaluate our Rep-Adapter on various settings to demonstrate its effectiveness, including many-shot and low-shot transfer learning with supervised and unsupervised ImageNet pre-training, few-shot transfer learning of CLIP, and semi-supervised learning.

### 4.1 Implementation Details

We implement the cross-domain transferring with the popular computer vision library VISSL (Goyal et al., 2021), which is inspired by the Visual Task Adaptation Benchmark (VTAB) (Zhai et al., 2020). VISSL contains 14 diverse classification downstream datasets, and two settings for tuning phase, 1) many shot transfer learning setting, where all training labels of the downstream tasks are used; 2) low-shot transfer learning setting that only trains on 1000 labeled images sampled from the downstream datasets. For few-shot learning with CLIP, we adopt 16-shot settings on 10 transfer learning datasets. See the supplementary for more details.

**Pre-Training Protocol.** We evaluate Rep-Adapter on self-supervised models, fully supervised models, and text-supervised models. We use two representative self-supervised methods, PIRL (Misra & Maaten, 2020), MoCoV2 (Chen et al., 2020b) on cross domain transfer learning and an additional self-supervised learning method SwAV (Caron et al., 2020) for semi-supervised learning on ImageNet, and use CLIP (Radford et al., 2021) for experiments with text-supervised pre-training. Please check the supplementary for more details.

**Baseline methods.** As our method aims to combine the advantage of the prevalent feature-based methods, *e.g.* linear probing, and fine-tuning. we choose these two method as our baseline methods. value for the classification layer, while fine-tuning is setting a same $\eta$ for all the network parameters.

### 4.2 Many-shot Transfer Learning

We first evaluate our method on many-shot transfer learning setting. We use all training data with labels from all 14 target datasets during the tuning phase. Results are presented in Tab. 3. Our Rep-Adapter generally outperforms the baseline methods, on 37 out of 42 scenarios. For example, our method improve over linear probing and fine-tuning by **16.1%** and **1.4%** with PIRL on average. It is worth mentioning that on the target datasets such as CIFAR-100, DTD, Flowers102, SUN397, Camelyon, Clevr-Dist, etc., Rep-Adapter surpasses the other two methods by a large margin whether using supervised or self-supervised pre-training models (*e.g.*, up to +46.7% on dSpr-Ori with PIRL, compared to linear probing).

---

[2]In Proposition 3.1, $\boldsymbol{\zeta}$ is fixed when updating $\boldsymbol{\omega}$. Therefore, the basic version of Rep-Adapter tuning is to update $\boldsymbol{\zeta}$ and $\boldsymbol{\omega}$ iteratively. To save the training cost, we further explore an approximated version of Rep-Adapter, by simultaneously updating $\boldsymbol{\zeta}$ and $\boldsymbol{\omega}$ in the same step. We empirically find that this version yields comparable results with the basic version (see Sec. 4.8 (b)). By default, we use the approximated version.

Table 3: **Results of many-shot and low-shot transfer learning.** We report the Top-1 Accuracy. The best and the second best results are marked with **bold** and underline.

| Dataset | | Caltech101 | CIFAR-100 | DTD | Flowers102 | Pets | SUN397 | SVHN | Camelyon | EuroSAT | Clevr-Count | Clevr-Dist | dSpr-Loc | dSpr-Ori | KITTI-Dist | Mean |
|---|---|---|---|---|---|---|---|---|---|---|---|---|---|---|---|---|
| *Many-shot Transfer Learning* | | | | | | | | | | | | | | | | |
| PIRL | Linear | 84.2 | 66.5 | 71.3 | 88.7 | 78.1 | 64.6 | 62.5 | 83.6 | 95.2 | 76.8 | 63.5 | 73.0 | 49.8 | 73.7 | 73.7 |
| | Fine-tuning | 85.4 | 81.8 | 67.7 | 95.2 | 85.7 | 67.9 | 95.7 | 87.4 | 97.7 | 99.4 | 93.7 | 100.0 | 96.0 | 84.2 | 88.4 |
| | Rep-Adapter | **88.0** | **83.7** | **72.5** | **95.6** | **88.3** | 70.7 | 96.5 | 88.6 | 98.0 | 99.9 | 94.1 | 100.0 | 96.5 | 85.3 | 89.8 |
| MoCoV2 | Linear | 87.1 | 70.9 | 72.4 | 91.7 | 78.4 | 65.9 | 70.9 | 84.1 | 95.8 | 77.7 | 67.7 | 83.3 | 55.2 | 74.3 | 76.8 |
| | Fine-tuning | 87.7 | 81.6 | 73.0 | 94.0 | **88.7** | 63.2 | 96.1 | 87.8 | 98.4 | 99.6 | 93.7 | 100.0 | 96.4 | 84.1 | 88.9 |
| | Rep-Adapter | 88.3 | 84.5 | 73.3 | 96.0 | 88.6 | 70.2 | 96.8 | 89.2 | 98.7 | 99.9 | 94.5 | 100.0 | 96.5 | 84.5 | 90.1 |
| Supervised | Linear | 88.5 | 73.6 | 71.2 | 85.7 | 91.5 | 66.0 | 69.3 | 85.4 | 95.4 | 65.7 | 62.0 | 81.8 | 60.9 | 72.2 | 76.4 |
| | Fine-tuning | 94.1 | 83.8 | 74.0 | 93.7 | 91.9 | 70.7 | 97.0 | 83.9 | 98.8 | 99.8 | 92.1 | 100.0 | 96.4 | 80.7 | 89.8 |
| | Rep-Adapter | 91.1 | 85.7 | 75.2 | 96.9 | 94.1 | 72.2 | 96.3 | 88.8 | 98.7 | 99.7 | 92.3 | 100.0 | 96.5 | 84.3 | 90.8 |
| *Low-shot Transfer Learning* | | | | | | | | | | | | | | | | |
| PIRL | Linear | 77.6 | 38.0 | 59.3 | 81.8 | 65.4 | 22.0 | 39.1 | 81.0 | 90.6 | 43.1 | 45.0 | 44.6 | 26.3 | 68.6 | 55.9 |
| | Fine-tuning | 74.0 | 45.9 | 55.5 | 89.3 | 71.4 | 19.9 | 88.0 | 85.4 | 94.5 | 52.0 | 59.7 | 79.8 | 48.2 | 70.4 | 66.7 |
| | Rep-Adapter | 78.9 | 45.7 | 61.3 | 89.2 | 76.0 | 20.1 | 88.2 | 86.4 | 95.7 | 56.9 | 62.5 | 86.4 | 47.8 | 70.6 | 69.0 |
| MoCoV2 | Linear | 79.3 | 44.7 | 64.1 | 85.8 | 67.4 | 25.5 | 46.0 | 83.3 | 92.2 | 46.7 | 50.5 | 53.0 | 27.0 | 68.9 | 59.6 |
| | Fine-tuning | 77.3 | 42.2 | 60.7 | 87.8 | 76.8 | 13.8 | 87.7 | 84.4 | 96.3 | 70.9 | 62.8 | 92.6 | 46.0 | 70.3 | 69.3 |
| | Rep-Adapter | 80.0 | 46.4 | 62.2 | 91.1 | 74.8 | 17.6 | 88.7 | 86.2 | 96.5 | 73.7 | 65.8 | 92.3 | 50.7 | 71.3 | 71.2 |
| Supervised | Linear | 84.9 | 53.1 | 62.6 | 87.7 | 89.9 | 30.9 | 43.7 | 80.9 | 90.8 | 38.9 | 42.0 | 42.5 | 29.2 | 66.0 | 60.2 |
| | Fine-tuning | 85.3 | 54.5 | 64.8 | 88.4 | 89.4 | 28.7 | 81.0 | 79.9 | 96.0 | 55.2 | 57.6 | 85.6 | 51.6 | 72.8 | 70.8 |
| | Rep-Adapter | 85.8 | 59.9 | 66.1 | 92.7 | 91.7 | 29.8 | 83.6 | 84.7 | 96.0 | 59.6 | 57.0 | 91.0 | 51.0 | 71.5 | 72.9 |

## 4.3 LOW-SHOT TRANSFER LEARNING

In the experiment of low-shot transfer learning, we follow the protocol in (Zhai et al., 2020) and only randomly reserve 1000 training data for each dataset. As shown in Tab. 3, the results show that Rep-Adatpter still outperforms the other two methods in general, with an average gain of **13.1%** and **2.3%** using PIRL pre-training. In addition, it can be seen from the experimental results that finetune performs better than linear probing in many-shot transfer learning, but in low-shot transfer learning, the conclusion is the opposite in some cases (*e.g.*, Caltech101 with PIRL and MoCoV2 and DTD with all the 3 self-supervised method).

## 4.4 COMPARISON WITH OTHER FINE-GRAINED MODEL ADAPTATION METHODS

In previous sections, we mainly compare our methods with the most representative fine-tuning and linear probing baselines. Here, we further compare with other fine-grained model adaptation methods. We first compare Rep-Adapter with LoRA (Hu et al., 2022) and Visual Prompt Tuning (Jia et al., 2022) on the low-shot (1000 examples) VTAB-1k benchmark with supervised pre-training weights. Then, we compare with the following fine-grained model adaptation methods in many-shot setting with supervised and PIRL pre-training weights: (a) $L^2$-SP (Li et al., 2018): using an $L^2$ penalty to encourage the fine-tuned network to be consistent with the pre-trained model on network weights, (b) DELTA (Li et al., 2019): using a penalty to encourage the fine-tuned network to

Table 4: Comparison with other model adaptation methods on low-shot & many-shot settings.

| Supervised VTAB-1k | Natural | Specialized | Structured | Mean |
|---|---|---|---|---|
| Fine-tuning | 70.3 | 77.6 | 42.7 | 60.2 |
| LoRA | 72.0 | 78.4 | 52.5 | 65.1 |
| Visual Prompt Tuning | 66.3 | 77.3 | 37.5 | 56.5 |
| Rep-Adapter | **72.8** | **80.6** | **56.7** | **67.6** |

| Supervised | CUBS | Stanford Cars | Flowers102 | WikiArt | Sketches | Mean |
|---|---|---|---|---|---|---|
| Linear | 74.07 | 70.81 | 85.67 | 61.60 | 75.50 | 73.53 |
| Fine-tuning | 81.67 | 89.34 | 93.67 | 73.28 | 77.93 | 83.18 |
| $L^2$-SP | 81.29 | 89.67 | 95.10 | 73.54 | 80.67 | 84.05 |
| DELTA | 82.40 | 89.50 | 95.33 | 74.99 | 79.40 | 84.32 |
| AutoLR | 81.78 | 89.10 | 94.06 | 76.34 | 80.32 | 84.32 |
| SpotTune | 84.03 | **92.40** | 96.34 | 75.77 | 80.20 | 85.75 |
| Rep-Adapter | **84.61** | 91.91 | **96.88** | **77.01** | 80.77 | **86.24** |

| PIRL | Caltech101 | CIFAR-100 | DTD | Flowers102 | Pets | Mean |
|---|---|---|---|---|---|---|
| Linear | 84.2 | 66.5 | 71.3 | 88.7 | 78.1 | 77.8 |
| Fine-tuning | 85.4 | 81.8 | 67.7 | 95.2 | 85.7 | 83.2 |
| $L^2$-SP | 91.9 | 78.8 | 68.7 | 93.7 | 85.2 | 83.6 |
| DELTA | collapse | collapse | collapse | 64.7 | collapse | - |
| AutoLR | **92.0** | 77.9 | 69.5 | 93.2 | 84.2 | 83.4 |
| Rep-Adapter | 88.6 | **83.7** | **72.5** | **95.6** | **88.3** | **85.7** |

be consistent with the pre-trained model on output features, (c) AutoLR (Ro & Choi, 2021): auto-tuning of layer-wise learning rates, (d) SpotTune (Guo et al., 2019): adaptively selecting between the fine-tuned and the pre-trained layers for each image.

**Low-shot.** As shown in Tab. 4, our method outperforms LoRA and Visual Prompt Tuning (VPT) by a large gap in low-shot setting. Remarkably, Rep-Adapter surpasses VPT by up to **19.15%** on structured datasets of VTAB-1k benchmark.

**Many-shot.** As shown in Tab. 4, our method clearly surpasses previous methods on many-shot settings. With both supervised and unsupervised (PIRL) pre-training, Rep-Adapter consistently

Table 5: **Top-1 Accuracy of Few-shot (16-shot) transfer learning with CLIP.** †: Zero-shot CLIP results are also listed for reference.

| Architecture | Method | Caltech101 | DTD | Flowers102 | Pets | SUN397 | Cars | Aircraft | Food101 | EuroSAT | UCF101 | Mean |
|---|---|---|---|---|---|---|---|---|---|---|---|---|
| **CLIP-R50** | Zero-shot† | 86.3 | 42.3 | 66.1 | 85.8 | 58.5 | 55.6 | 17.3 | 77.3 | 37.6 | 61.5 | 58.8 |
| | CoOp | 91.8 | 63.6 | 94.5 | 87.0 | 69.3 | 73.4 | 31.3 | 74.7 | 83.5 | 75.7 | 74.5 |
| | Tip-Adapter | 90.9 | 60.9 | 89.9 | 88.1 | 66.9 | 66.8 | 29.8 | 77.8 | 70.5 | 70.6 | 71.2 |
| | Tip-Adapter-F | 93.0 | 66.6 | 94.8 | **89.7** | 71.5 | 75.7 | 35.6 | **79.4** | 84.5 | 78.0 | 76.9 |
| | Rep-Adapter | **93.9** | **68.0** | **95.5** | 89.6 | **71.8** | 76.4 | 36.2 | 79.0 | **85.9** | 79.3 | **77.6** |
| **CLIP-ViT-B/32** | Zero-shot† | 90.9 | 44.0 | 67.0 | 87.5 | 61.9 | 60.6 | 19.2 | 80.5 | 45.2 | 62.0 | 61.9 |
| | CoOp | 94.6 | 65.4 | 95.0 | 88.7 | 72.4 | 76.1 | 33.2 | 78.5 | 83.4 | 78.7 | 76.6 |
| | Rep-Adapter | **95.4** | **67.1** | **95.7** | **89.8** | **73.5** | **77.4** | **36.3** | **81.3** | **85.8** | **80.0** | **78.2** |

outperforms $L^2$-SP, DELTA, AutoLR by a large margin. Our method can even outperforms the dynamic model adaptation method, SpotTune, by **0.49%** on average. Note that SpotTune uses an additional policy network, and has *more than 2× model size* during inference. In contrast, the improvement of Rep-Adapter comes for free.

## 4.5 FEW-SHOT TRANSFER LEARNING WITH CLIP

To further evaluate the effectiveness of our Rep-Adapter, we conduct experiments of few-shot (16-shot) transfer learning with CLIP (Radford et al., 2021), and compare the results with Zero-shot CLIP (Radford et al., 2021), CoOp (Zhou et al., 2022b), Tip-Adapter (Zhang et al., 2022). For CLIP models, we only apply our Rep-Adapter on the visual encoder. In addition to CLIP-ResNet50, we also apply our method to CLIP-ViT-B/32 to explore its generalization ability on transformer. For transformers, we apply Rep-Adapter on every fully-connected layers. More implementation details can be found in the supplementary material.

As shown in Tab. 5, Rep-Adapter generally outperforms previous methods. On average, Rep-Adapter improves over CoOp (Zhou et al., 2022b) by 3.1% with ResNet50 and by 1.6% with ViT-B/32. It is noteworthy that Rep-Adapter does not introduce any extra model parameter during inference, in contrast to prompt tuning methods and previous adapter-based methods. These results suggest that Rep-Adapter is effective on vision-language pretrained models and vision transformers.

## 4.6 SEMI-SUPERVISED IMAGENET

Models pretrained with self-supervised learning achieves strong performance when adapted with small fraction of labeled data. Following the semi-supervised protocol of (Chen et al., 2020a), we conduct experiments on the same fixed splits, 1% and 10% labeled ImageNet training data. We use models pretrained with self-supervised methods PIRL (Misra & Maaten, 2020), MoCoV2 (Chen et al., 2020b)

Table 6: **Results on semi-supervised Image-Net.**

| Method | Protocol | Top-1 (%) Label fraction | | Top-5 (%) Label fraction | |
|---|---|---|---|---|---|
| | | 1% | 10% | 1% | 10% |
| PIRL | Linear | 35.0 | 49.9 | 61.1 | 75.7 |
| | Fine-tuning | 40.2 | 62.4 | 67.6 | 85.0 |
| | Rep-Adapter | **41.5** | **63.5** | **69.2** | **86.1** |
| MoCoV2 | Linear | 24.8 | 40.3 | 53.1 | 69.4 |
| | Fine-tuning | 39.1 | 61.8 | 68.3 | 85.1 |
| | Rep-Adapter | **43.4** | **64.0** | **72.5** | **86.6** |
| SwAV | Linear | 51.4 | 65.5 | 77.4 | 87.0 |
| | Fine-tuning | 51.9 | 69.6 | 78.2 | **89.8** |
| | Rep-Adapter | **52.6** | **70.0** | **79.0** | **89.8** |

and SwAV (Caron et al., 2020). As shown in Tab. 6, Rep-Adapter consistently outperforms the other two generally used methods. On MoCoV2, Rep-Adapter improves over the standard fine-tuning by **4.3%** and **2.2%** with 1% and 10% label fraction, respectively.

## 4.7 OBJECT DETECTION & INSTANCE SEGMENTATION

To further explore the transferability of Rep-Adapter, we evaluate it on object detection and instance segmentation. We use Faster-RCNN (Ren et al., 2015) on PASCAL VOC (Everingham et al., 2010) dataset and Mask-RCNN (He et al., 2017) on Cityscapes (Cordts et al., 2016) and COCO 2017 (Lin et al., 2014) datasets. We report the bounding box mAP ($AP^{Box}$) and mask mAP

Table 7: **Results on object detection and instance segmentation.**

| (F: Fine-tuning, R: Rep-Adapter) | | PIRL F | R | MoCoV2 F | R | SwAV F | R | Supervised F | R |
|---|---|---|---|---|---|---|---|---|---|
| **VOC** | $AP^{Box}$ | 52.3 | **56.3** | 53.8 | **56.4** | 52.9 | **55.8** | 53.8 | **55.6** |
| **Cityscapes** | $AP^{Mask}$ | 33.3 | **36.0** | 33.9 | **37.3** | 35.3 | **38.2** | 33.2 | **35.4** |
| **COCO** | $AP^{Box}$ | 37.9 | **38.9** | 38.8 | **39.4** | 40.0 | **40.3** | 39.0 | **39.5** |
| | $AP^{Mask}$ | 34.5 | **35.7** | 35.4 | **36.0** | 36.7 | **37.0** | 35.6 | **36.2** |

($AP^{Mask}$) in Tab. 7. Rep-Adapter consistently outperforms the generally used fine-tuning (by up to **4.0 AP** on VOC). Besides, these results also demonstrate that our method is not limited by the batch size, because these experiments are performed with extremely low batch size (2 on each GPU).

Table 8: **Ablation Study.** A: Effect of the scaling factor $\delta$; B: Effect of joint optimization of $\delta, \zeta$ with $\omega$.; C: Effect of the batch-wise statistics $\sigma, \mu$; D: Effect of the BN position; E: Effect of $\omega_0$ weight initialization; F: Effect of weight frozen in frozen branch. Results are shown in "mean (std)" of multiple trial results.

| Study on | Parameters init w/ $\omega_0$ | BN learnable $\gamma, \beta$ | BN iteratively trained with $\omega$ | BN frozen $\sigma, \mu$ | BN in branch | Conv frozen | Caltech101 | CIFAR-100 | DTD | Flowers102 | Pets |
|---|---|---|---|---|---|---|---|---|---|---|---|
| Ours | ✓ | ✓ | ✗ | ✗ | ✓ | ✓ | **84.7 (1.4)** | **59.6 (0.2)** | **64.2 (2.4)** | **92.7 (0.3)** | **90.3 (2.6)** |
| **(a)** BN $\delta$ | ✓ | ✗ | ✗ | ✗ | ✓ | ✓ | 83.8 (1.4) | **59.6 (0.1)** | 63.1 (3.0) | 92.5 (0.2) | 87.1 (2.8) |
| **(b)** BN $\delta, \zeta$ | ✓ | ✓ | ✓ | ✗ | ✓ | ✓ | 84.2 (1.6) | 59.5 (0.2) | 63.8 (2.8) | 92.5 (0.4) | 90.0 (2.5) |
| **(c)** BN $\sigma, \mu$ | ✓ | ✓ | ✗ | ✓ | ✓ | ✓ | 82.5 (2.2) | 59.3 (0.4) | 61.9 (0.7) | 92.3 (0.2) | 88.1 (2.0) |
| **(d)** BN Pre-add. | ✓ | ✓ | ✗ | ✗ | ✗ | ✓ | 82.1 (3.5) | **59.6 (0.2)** | 61.6 (3.1) | 91.8 (0.4) | 85.9 (5.4) |
| **(e)** Weight init. | ✗ | ✓ | ✗ | ✗ | ✓ | ✓ | 26.6 (6.2) | 11.2 (0.9) | 18.4 (3.1) | 39.7 (8.3) | 17.0 (1.1) |
| **(f)** Weight frozen | ✓ | ✓ | ✗ | ✗ | ✓ | ✗ | 81.8 (3.6) | 58.9 (0.2) | 60.9 (3.4) | 91.8 (0.7) | 85.5 (5.3) |
| Fine-tuning | ✓ | ✓ | ✗ | ✗ | - | ✗ | 81.6 (5.1) | 57.8 (1.6) | 60.2 (5.4) | 91.5 (1.6) | 83.5 (7.7) |

## 4.8 ABLATION STUDY

In this section, we perform ablation study on different components of Rep-Adapter.

**(a) Effect of the scaling factor $\delta$.** As learnable weights $\gamma$ of BN is used as the scaling factor in our implementation, the effect of the scaling factor $\delta$ of the frozen branch can be ablated by frozen $\gamma$ of the BN in the frozen branch. As shown in Tab. 8, enabling learnable weight of BN, *i.e.*, learnable $\delta$ in Rep-Adapter, slightly increase network performance on all the five datasets. Thus, we enabling it be default, though it does not affect the simulated learning rate of Rep-Adapter.

**(b) Effect of joint optimization of $\delta, \zeta$ with $\omega$.** In Sec. 3.3, we discussed Rep-Adapter under the situation when the scaling factors $\delta, \zeta$ of the frozen branch and the fine-tuning branch are fixed when updating $\omega$, *i.e.*, $\delta, \zeta$ and network parameter $\omega$ are updated iteratively in separate steps. To save the training cost and simplify the pipeline, we jointly optimize $\delta, \zeta$ and $\omega$ in each single step as in Equation (12). As shown in Tab. 8, joint and iterative optimization achieve comparable performance.

**(c) Effect of the batch-wise statistics $\sigma, \mu$.** In Sec. 3.3, we discussed Rep-Adapter on linear case, which covers using BN with running statistics $\sigma, \mu$. However, its a common practice in re-parameterization literature to enable the batch-wise statistics in BN during training (Ding et al., 2021c;b). Here, we explore how will the non-linearity and stochasticity brought by batch-wise statistics affect the performance. As shown in Tab. 8, enabling the batch-wise statistics (ours) consistently outperforms the case using frozen $\sigma, \mu$.

**(d) Effect of the BN position.** In Rep-Adapter, we use BN in both branches, before adding the output of the branches together. Alternatively, the BN can also be used after addition, where only Conv layers are in the branches. The results of the two cases is shown in Tab. 8. Using BN in the branches clearly improves the performance of Rep-Adapter.

**(e) Effect of $\omega_0$ weight initialization.** In Rep-Adapter, the weight of the frozen branch is initialized with the pretrained weight $\omega_0$. As shown in Tab. 8, random initializing the frozen branch would drastically damage the performance of Rep-Adapter (*e.g.*, from 90.3% to 17.0% on Pets), demonstrating that Rep-Adapter *do* make use of the information in the frozen branch and the importance of pretrained weight initialization in the frozen branch.

**(f) Effect of weight frozen in frozen branch.** Tab. 8 also shows the result of not freezing the weight in the frozen branch of Rep-Adapter. The performance drop considerably on all the five datasets (up to 4.8% on Pets) when making the frozen branch learnable, and is comparable to fine-tuning baseline. This demonstrate that the performance gain achieved by Rep-Adapter over fine-tuning is *not* caused by re-parameterization and *not* by increasing the network complexity during training. The frozen weights are critical for Rep-Adapter to achieve the performance improvement.

## 5 CONCLUSION

Model adaptation is a long-standing problem in transfer learning. We notice the fact that most previous model adaptation baselines have several downsides, such as tedious manual tuning of learning rate and extra computational cost during inference phase. To the best of our knowledge, we are the first to formulate automatic model adaptation into a bi-level optimization problem. We propose a parameter-free protocol that can jointly optimize the learning rate and model parameters during tuning phase with a theoretical basis. Our experiments empirically evidence the effectiveness and generalization ability of our method. We hope our work can benefit future works on model adaptation and reduce the human effort to manually tuning the hyper-parameters during transfer learning.

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

## A  APPENDIX

### A.1  A SIMPLIFIED CASE OF PROPOSITION 3.1

**Proposition A.1.** *Given a pre-trained linear network layer $\psi(\boldsymbol{\omega}^0)$. Let $\psi^{\mathcal{R}}(\boldsymbol{x}) = \psi(\boldsymbol{\omega}^0) + diag(\boldsymbol{\zeta})\psi(\boldsymbol{\omega}^{\mathcal{R}})$ denotes this layer under adapter tuning scheme at learning rate $\eta_m$, and $\psi^{\mathcal{F}}(\boldsymbol{\omega})$ denotes this layer with each of its filters fine-tuning at a certain learning rate. Suppose the network functions of the two schemes are identical at tuning step $t$, for one of its filters, fine-tuning at a certain learning rate $\eta \in \mathbb{R}$, then there always exists scaling factor $\zeta \in \mathbb{R}$ for this filter such that after one step of tuning, $\psi^{\mathcal{F}}(\boldsymbol{x}) = \psi^{\mathcal{R}}(\boldsymbol{x})$ still holds.*

*Proof.* Let $\boldsymbol{u}$ be one of the filters of $\psi^{\mathcal{F}}(\boldsymbol{x})$, $\boldsymbol{u}_0$ be the corresponding filter of the pre-trained weight, and $\boldsymbol{v}$ be the corresponding filter of $\psi^{\mathcal{R}}(\boldsymbol{x})$. Suppose the network functions of the two schemes are identical at step $t$. We have $\psi_t^{\mathcal{F}}(\boldsymbol{x}) = \psi_t^{\mathcal{R}}(\boldsymbol{x})$ and $\partial\mathcal{L}_t/\partial\psi_t^{\mathcal{R}} = \partial\mathcal{L}_t/\partial\psi_t^{\mathcal{F}}$. First, we calculate the weight change of $\boldsymbol{v}$ in the adapter:

$$\Delta\boldsymbol{v}_t = -\eta_m\frac{\partial\mathcal{L}_t}{\partial\boldsymbol{v}_t} = -\eta_m\zeta\frac{\partial\mathcal{L}_t}{\partial\psi_t^{\mathcal{R}}}. \tag{13}$$

Let $\widetilde{\boldsymbol{u}}_t = \boldsymbol{u}_0 + \zeta\boldsymbol{v}_t$, this layer function after one-step adapter tuning is:

$$\begin{aligned}
\psi_{t+1}^{\mathcal{R}}(\boldsymbol{x}) &= \boldsymbol{u}_0\boldsymbol{x} + \zeta(\boldsymbol{v} + \Delta\boldsymbol{v})\boldsymbol{x} \\
&= (\widetilde{\boldsymbol{u}}_t - \eta_m\zeta\frac{\partial\mathcal{L}_t}{\partial\psi_t^{\mathcal{R}}})\boldsymbol{x}.
\end{aligned} \tag{14}$$

As for the fine-tuned case:

$$\psi_{t+1}^{\mathcal{F}}(\boldsymbol{x}) = \boldsymbol{u}_{t+1}\boldsymbol{x} = (\boldsymbol{u}_t - \eta\frac{\partial\mathcal{L}_t}{\partial\psi_t^{\mathcal{F}}})\boldsymbol{x}. \tag{15}$$

Recall that $\psi_t^{\mathcal{F}}(\boldsymbol{x}) = \psi_t^{\mathcal{R}}(\boldsymbol{x})$, and $\partial\mathcal{L}_t/\partial\psi_t^{\mathcal{R}} = \partial\mathcal{L}_t/\partial\psi_t^{\mathcal{F}}$. Thus, there exists:

$$\boldsymbol{\zeta} = \sqrt{\frac{\eta}{\eta_m}}, \quad \text{such that} \ \ \psi_{t+1}^{\mathcal{F}}(\boldsymbol{x}) = \psi_{t+1}^{\mathcal{R}}(\boldsymbol{x}), \tag{16}$$

which proves that after one step of tuning, $\psi^{\mathcal{F}}(\boldsymbol{x}) = \psi^{\mathcal{R}}(\boldsymbol{x})$ still holds. $\qquad\square$

## A.2 IMPLEMENTATION DETAILS

**Datasets.** For full-shot and few-shot learning experiments, we perform all of pre-training on ImageNet (Krizhevsky et al., 2012), a benchmark for image classification including around 128 million training images from 1,000 categories. We adopt 14 Visual Task Adaptation Benchmark (VTAB) tasks which cover a broad spectrum of domains and semantics, containing object identification (Caltech101 (Fei-Fei et al., 2006), CIFAR-100 (Krizhevsky et al., 2009), Flowers102 (Nilsback & Zisserman, 2008), and Pets (Parkhi et al., 2012)), texture classification (DTD (Cimpoi et al., 2014)), scene classification (SUN397 (Xiao et al., 2010), and SVHN (Netzer et al., 2011)), pathology detection (Patch Camelyon (Veeling et al., 2018)), satellite image classification (EuroSAT (Helber et al., 2019)), counting(Clevr/count (Johnson et al., 2017)), localization (dSprites/location (Matthey et al., 2017)), orientation (dSprites/orientation (Matthey et al., 2017)), and 3D geometry (Clevr/distance (Johnson et al., 2017), and KITTI-Dist (Geiger et al., 2013)). We set the same train/test splits as (Zhai et al., 2020) and also use the regular test sets for the low-shot (1000 examples) tasks.

For CLIP transfer learning, we benchmark on 10 publicly available image classification datasets used in CLIP: Caltech101 (Fei-Fei et al., 2006), OxfordPets (Parkhi et al., 2012), StanfordCars (Krause et al., 2013), Flowers102 (Nilsback & Zisserman, 2008), Food101 (Bossard et al., 2014), FGVCAircraft (Maji et al., 2013), SUN397 (Xiao et al., 2010), DTD (Cimpoi et al., 2014), EuroSAT (Helber et al., 2019) and UCF101 (Soomro et al., 2012).

We provide the licenses of the datasets mentioned above if known. ImageNet is available for free to researchers for non-commercial use. Pets is under the CC BY-SA 4.0 license. DTD is made available to the computer vision community for research purposes. SVHN is for non-commercial use only. Patch Camelyon is under the CC0 license. Clevr/count and Clevr/distance are under the CC BY 4.0 license. KITTI-Dist is under the CC BY-NC-SA 3.0 license. The datasets we use are publicly available, so that we obtain consent to use them by default. These datasets do not contain personally identifiable information or offensive content.

**Self-supervised training protocols.** We adopt three self-supervised training protocols to pre-train the model in our experiment section, here are the details of each method.

- **PIRL** levages pre-tasks, learning representations that are invariant to the transformations and retaining semantic information;

- **MoCoV2** establishes a strong contrastive unsupervised learning baseline with a memory bank to restore negative samples.

- **SwAV** simultaneously clusters the data while enforcing consistency between cluster assignments produced for different augmentations of the same image, instead of comparing features directly as in contrastive learning.

**Training Settings.** For the VTAB downstream tasks, we set the total batch size of 256 on 8 NVIDIA GTX 2080Ti GPUs and use SGD with momentum of 0.9 without weight decay. We resize all images to $224 \times 224$. We decay the learning rate by a factor of 10 after $\frac{1}{3}$ and $\frac{2}{3}$ of the training time. We train 2500 training steps for low-shot (1000 examples) transfer tasks. For each many-shot transfer task we try {2500, 5000, 10000} training steps and select the best one.

For CLIP transfer learning, we only apply Rep-Adapter on the visual encoder. We use AdamW optimizer and cosine learning rate scheduler. We train the model for 20 epochs on all the datasets except EuroSAT, for which we use 100 epochs.

For semi-supervised ImageNet tasks with self-supervised pretrained models, we train for 20 epochs with a total batch size of 256 on 4 NVIDIA GTX 2080Ti GPUs and use SGD optimizer with a momentum of 0.9, and a weight decay of 5e-4. The learning rate decays by 0.2 at 12 and 16 epochs. Only random resize&crop to $224 \times 224$, horizontal flip are used for data augmentation.

For object detection and instance segmentation downstream tasks, we train with the FPN (Lin et al., 2017) backbone on 8 NVIDIA GTX 2080Ti GPUs, follow the training settings in MMSelfSup (Contributors, 2021) and implement in Detectron2 (Wu et al., 2019) library.

Code will be released.

## A.3 Additional Related Work

**Model Adaptation.** *Fine-tuning* (Girshick et al., 2014; Yosinski et al., 2014) has achieved *state-of-the-art* performance on many computer vision tasks and NLP tasks, including image classification (Kornblith et al., 2019; Hermans et al., 2017), object detection (Girshick et al., 2014), semantic segmentation (Long et al., 2015), text classification (Wang et al., 2019) and question answering (Rajpurkar et al., 2016). *Adapter tuning* (Houlsby et al., 2019; Yuan et al., 2020) adds light-weight modules on the pre-trained network to adapt the model without changing its original parameters. Clip-adapter (Gao et al., 2023) and Tip-adapter (Zhang et al., 2022) adopt residual adapter modules for few-shot transfer learning of CLIP (Radford et al., 2021). Similarly, *prompt tuning* (Lester et al., 2021) in NLP field learns input prompt tokens for a frozen Transformer network to achieve transfer learning. Prompt tuning is proved to perform better than fine-tuning in few-shot scenarios (Gao et al., 2021). Recently, CoOp (Zhou et al., 2022b) applies prompt tuning to improve the few-shot transfer learning performance of visual-linguistic pre-trained models, *e.g.*, CLIP (Radford et al., 2021). Uni-perceiver (Zhu et al., 2022) uses prompt tuning for few-shot transfer of a generic multi-modal perception architecture. Visual Prompt Tuning (Jia et al., 2022) uses prompt tuning for visual transfer learning tasks.

**Other AutoML works related to transfer learning.** Apart from the works we introduced in the main text, there exists a few other works on transfer learning that are relevant to AutoML. AutoFreeze (Liu et al., 2021) accelerates fine-tuning by automatically freezing layers. L2TL (Zhu et al., 2020) jointly trains the model on source and target dataset and uses adaptive weights to balance between losses. Other automated methods using a new target network include L2T-ww (Jang et al., 2019).

## A.4 Experiments on Larger Backbones

To evaluate Rep-Adapter on larger backbones, we performs experiments with ResNet101 on supervised, many-shot settings. The results are shown in Table 9. Comparing to ResNet50, Rep-Adapter with ResNet101 achieves higher performance (+0.57%). It achieves remarkable performance improvement comparing to previous transfer learning method. For example, Rep-Adapter with ResNet101 outperforms fine-tuning by 1.87% on average.

Table 9: **Results of ResNet101 with supervised pre-training, on many-shot settings.**

|  | CUBS | Stanford Cars | Flowers | WikiArt | Sketchs | Mean |
|---|---|---|---|---|---|---|
| Fine-tuning | 84.26 | **92.13** | 94.80 | 73.14 | 80.37 | 84.94 |
| $L^2$-SP (Li et al., 2018) | 82.27 | 90.50 | 94.36 | 74.28 | 81.06 | 84.49 |
| DELTA (Li et al., 2019) | 84.05 | 91.47 | 95.09 | 75.63 | 81.64 | 85.58 |
| Rep-Adapter | **85.05** | 91.88 | **97.03** | **78.30** | **81.81** | **86.81** |

## A.5 Additional Ablation Study

**Ablation on scaling factors.** The scaling factors are the crucial components of the proposed Rep-Adapter.However, using scaling factors without other components in Rep-Adapter (two branches, BN in the branch) or using Rep-Adapter without scaling factors are both inferior to our proposed Rep-Adapter. We have performed Ablation study by using scaling factors without other components in Rep-Adapter to demonstrate the effectiveness. The results are shown in As shown in Tab. 10,. The result of w/o scaling and w/o BN is the result of case (a) and (d) in Tab. 8. By comparing the results of Rep-Adapter with (b) w/o scaling, we can see that scaling is an important component of Rep-Adapter. Results of (a) only scaling shows that the scaling needs the other design of Rep-Adapter to be effective. Results of (c) w/o BN shows that the BN in the branch is also an important factor to achieve good results with Rep-Adapter.

**Ablation on initialization of the tuning branch.** We perform experiment by randomly initializing. As shown in Tab. 10, random initializing the tuning branch would clearly damage the performance of Rep-Adapter, demonstrating that the pretrained initialization is important to the tuning branch.

Table 10: **Additional Ablation Study.** (a):Only scaling factor, without other architecture of Rep-Adapter; (b) Rep-Adapter without scaling factor; (c) BN not included in Rep-Adapter; (d) Random initialize frozen branch; (e) Random initialize tuning branch. Results are shown in "mean (std)" of multiple trial results.

| Study on | Caltech101 | CIFAR-100 | DTD | Flowers102 | Pets |
|---|---|---|---|---|---|
| Ours | **84.7 (1.4)** | **59.6 (0.2)** | **64.2 (2.4)** | **92.7 (0.3)** | **90.3 (2.6)** |
| **(a)** Only scaling | 81.7 (4.4) | 58.1 (1.6) | 60.5 (3.1) | 91.3 (1.3) | 84.0 (5.1) |
| **(b)** w/o scaling | 83.8 (1.4) | **59.6 (0.1)** | 63.1 (3.0) | 92.5 (0.2) | 87.1 (2.8) |
| **(c)** w/o BN | 82.1 (3.5) | **59.6 (0.2)** | 61.6 (3.1) | 91.8 (0.4) | 85.9 (5.4) |
| **(d)** Random init. F. | 26.6 (6.2) | 11.2 (0.9) | 18.4 (3.1) | 39.7 (8.3) | 17.0 (1.1) |
| **(e)** Random init. T. | 74.8 (5.9) | 52.5 (1.4) | 53.9 (3.6) | 88.1 (1.1) | 79.1 (4.5) |
| Fine-tuning | 81.6 (5.1) | 57.8 (1.6) | 60.2 (5.4) | 91.5 (1.6) | 83.5 (7.7) |

## A.6 ADDITIONAL ANALYSIS ON EXTREME CASE

We noticed that our model is inferior to linear probing on a specific scenario, low-shot SUN397, with less than 3 samples per class. Although our Rep-Adapter has the theoretical potential to decay to complete linear probing, this may not hold in this extreme case, as the scaling factor $\eta$ simulating learning rate may also over-fit. We provide an additional exploration on this dataset. We select the pre-trained protocol, SwAV, where our method has a significant gap between linear probing. We increase the number of sampled images to repeat the low-shot experiment, and report the results in Table 11. We can observe our method performance increases while the images increase, consistently outperforms fine-tuning. When using 7940 traing samples (20 per class), our method performs comparably with linear probing and eventually surpasses it when sampling 40 images per class.

Table 11: **Increasing number of sampled images on SUN397.** The gap between our method and linear probing shrinks while sampling number increases, and eventually surpasses it.

| Samples | 1000 | 3970 | 7940 | 11910 | 15880 |
|---|---|---|---|---|---|
| Linear | **32.2** | **51.5** | **54.1** | **58.9** | 60.1 |
| Fine-tuning | 22.4 | 45.3 | 51.1 | 56.9 | 57.9 |
| Rep-Adapter | 24.3 [- 7.9] | 46.6 [- 4.9] | 53.6 [- 0.5] | 58.6 [- 0.3] | **60.40** [+ 0.3] |

