# OpenReview forum: "Rep-Adapter: Parameter-free Automatic Adaptation of Pre-trained ConvNets via Re-parameterization"
_ICLR.cc/2024/Conference — Submitted to ICLR 2024_

### Official Review · Reviewer_xJSa · 2023-10-31

**Soundness:** 2 fair
**Presentation:** 2 fair
**Contribution:** 2 fair
**Rating:** 3
**Confidence:** 5

**Summary:**

This paper proposed an adaption tuning method for ConvNets. The learnable parameters of can be re-parameterized to the original conv module to achieve parameter-free adaption. On different pre-trained models, the authors demonstrate the proposed method can achieve good performance.

**Strengths:**

1. The proposed method indeed provide some interesting experiments for parameter-free adaption of ConvNets, especially on few-shot setting,

**Weaknesses:**

1. The proposed method has limited novelty. A similar adaption and re-parameterization method has been proposed in earlier methods. [1]
2. The proven "layers tuning at any learning rate via Rep-Adapter" does not make too much sense. Adding scaling factors is equivalent to the adaptive learning rate, and has nothing to do with the proposed Rep-Adapter.
3. While the proposed method can be re-parameterized after training, the modules equivalent to the original network size still need to be fully tuned during training. I don't see any advantage of this compared to parameter-efficient tuning methods except the performance gain reported in the paper.
4. The reported results are not verified using different random seeds. No error bar is reported.


[1] Sylvestre-Alvise Rebuffi et al. Efficient parametrization of multi-domain deep neural networks. CVPR 2018.

**Questions:**

1. Can you explain why Rep-Adapter outperforms the full fine-tuning in Table 3?  The details of the initial learning rate for fine-tuning and rep-adapter are missing. Also, does full fine-tuning conduct the same number of epochs as the Rep-Adapter?  10000 steps seem not enough for fine-tuning, can you justify it?
2. For ablation study (e), it would be of more interest to see different initialization of $w_{R}$ rather than $w_0$.

---

> ### Author Response · Authors · 2023-11-22
> **Author response to Reviewer xJSa (Part 1)**
>
> Thank you for your detailed and constructive comments. We respond to all the issues you pointed out in details below. We hope our response and rebuttal revision will address your concerns.
>
> **Weakness 1. Novelty over [1].**
>
> Thank you for pointing out the related literature! We would like to summarize our novelty over this work as below.
>
> 1. Technical difference. In [1], the authors propose an adapter to modify the “diagonal” elements of the filters additively. In this paper, we propose an adapter to modify the learning rate of every element of the filters.
> 2. Structure difference. In [1], the adapter is simply a scaling layer in parallel with the convolutional layer. Different from this, our Rep-Adapter contains a tuning branch with a new convolutional layer with two scaling layers in each branch to balance the learning rate.
> 3. Re-parameterization difference. Different from [1], Rep-Adapter also optimizes the pre-trained model in a larger network capacity with non-linearity BN in the branch. This is similar to the advantage of structural re-parameterization training over traditional training and has been generally proved by the success of ACNet (Ding et al., 2019), RepVGG (Ding et al., 2021c), etc. This advantage of structural re-parameterization can also be proved by our ablation study (c), where we study the important component of structural re-parameterization (Ding et al., 2021c;b), the batch-wise statistics in BN during training.
>
>
> **Weakness 2. Adding scaling factors is equivalent to the adaptive learning rate, and has nothing to do with the proposed Rep-Adapter.**
>
> Thank you for this comment. The scaling factors are the crucial components of the proposed Rep-Adapter. As shown by Proposition 3.1, the optimization of learning can be relaxed to the optimization of the scaling factors. However, using scaling factors without other components in Rep-Adapter (two branches, BN in the branch) or using Rep-Adapter without scaling factors are both inferior to our proposed Rep-Adapter. We have performed an Ablation study by using scaling factors without other components in Rep-Adapter to demonstrate the effectiveness. The results are shown in the Table below. The result of w/o scaling and w/o BN is the result of cases (a) and (d) in Table 8. By comparing the results of Rep-Adapter with (b) w/o scaling, we can see that scaling is an important component of Rep-Adapter. Results of (a) only scaling shows that the scaling needs the other design of Rep-Adapter to be effective. Results of (c) w/o BN show that the BN in the branch is also an important factor to achieve good results with Rep-Adapter.
>
>
> | Method | Caltech101 | CIFAR-100 | DTD | Flowers102 | Pets|
> | :---|:----:| :----: | :----: | :----: | :----: |
> | Rep-Adapter| 84.7 (1.4)  | 59.6 (0.2) | 64.2 (2.4) | 92.7 (0.3) | 90.3 (2.6)|
> | (a) Only scaling| 81.7 (4.4)  | 58.1 (1.6) | 60.5 (3.1) | 91.3 (1.3) | 84.0 (5.1)|
> | (b) w/o scaling| 83.8 (1.4)  | 59.6 (0.1) | 63.1 (3.0) | 92.5 (0.2) | 87.1 (2.8)|
> | (c) w/o BN| 82.1 (3.5)  | 59.6 (0.2) | 61.6 (3.1) |  91.8 (0.4) | 85.9 (5.4) |
> | Fine-tuning| 81.6 (5.1)  | 57.8 (1.6) | 60.2 (5.4) | 91.5 (1.6) | 83.5 (7.7)|
>
> Therefore, scaling factors and other designs of Rep-Adapter are closely related and rely on each other. We have added these results and analysis to Appendix A.5 of the rebuttal revision (marked by purple).

---

> ### Author Response · Authors · 2023-11-22
> **Author response to Reviewer xJSa (Part 2)**
>
> **Weakness 3. Advantage over parameter-efficient tuning methods.**
>
> Thank you for this comment. We would like to address this concern from four aspects.
>
> 1. Hyper-parameter tuning. Parameter-efficient tuning methods usually require a higher learning rate than fine-tuning and could suffer from tedious Hyper-parameter search. Our Rep-Adapter can automatically adjust the learning rate to avoid manual hyper-parameter search. Remarkably, we use one default learning rate for Rep-Adapter on all the datasets in experiments.
>
> 2. We would like to point out that our method does not aim at fast adaptation, but aims at the transfer learning performance, which is in line with L2-SP, DELTA, AutoLR, and SpotTune. Also, previous structural re-parameterization works usually achieve performance improvements at the cost of training efficiency. Compared to AutoLR (Ro & Choi, 2021), Rep-Adapter trains 1.67× faster and achieves 2.3% performance improvement on average of 4 different datasets with PIRL (Table 5).
>
> 3. Rep-Adapter achieves significant performance improvements over parameter-efficient tuning methods with minor differences in training speed. For example, Rep-Adapter brings 11.1% mean top-1 accuracy improvements on supervised VTAB-1k compared to a parameter-efficient tuning method, Visual Prompt Tuning (Table 4 in the paper).
>
> 4. We would also like to point out that most other parameter-efficient tuning methods, such as Tip-Adapter and VPT, could not achieve fast adaptation, as the gradient of all the parameters still needs to be calculated in the back-propagation and saved for the update of that small set of parameters. This issue has also been pointed out in previous literature [*1].
>
> [*1] Sung, Yi-Lin, et al. ``LST: Ladder Side-Tuning for Parameter and Memory Efficient Transfer Learning." NeurIPS, 2022.
>
> In summary, Rep-Adapter significantly outperforms parameter-efficient tuning methods with minor difference in training cost and does not need tedious Hyper-parameter search. We hope these advantages over parameter-efficient tuning methods could address your concern.
>
> **Weakness 4. Results are not reported with deviation.**
>
> Thank you! Sorry for the confusion. For clarity, we only report the results in the ablation study with standard deviation. In fact, all of our results reported in the paper are the mean results over 3 independent runs using different random seeds. We will clarify this in the section on experimental details.
>
> We have calculated the standard deviation of the numbers over 3 independent runs of the few-shot transfer learning experiments. When taking the variance into account, Rep-Adapter still achieves clear improvements over other methods. For example, on Caltech101 dataset, the standard deviation is 0.4, which is much smaller than the gap between the results of Rep-Adapter and baseline methods (3% improvements over Tip-Adapter). We will add the standard deviation of all our results in the paper.

---

> ### Author Response · Authors · 2023-11-22
> **Author response to Reviewer xJSa (Part 3)**
>
> **Question 1. Why Rep-Adapter outperforms the full fine-tuning in Table 3? Training details?**
>
> Thanks for this valuable question. Rep-Adapter outperforms full fine-tuning due to the following reasons.
>
> 1. During Rep-Adapter tuning, every filter of ConvNets can be seen as tuning at a different learning rate, while in fine-tuning, every filter uses the same predefined learning rate, which could be sub-optimal as early layers may need a lower learning rate than later layers.
>
> 2. Compared to fine-tuning, our Rep-Adapter automatically finds the optimal transfer learning protocol that saves the tediously searching for hyper-parameter settings.
>
> 3. Rep-Adapter also benefits from the structural re-parameterization design of Rep-Adapter. Structural re-parameterization eases network optimization and improves network performance by increasing network capacity and complexity during training. This advantage of structural re-parameterization has been generally proved by the success of ACNet (Ding et al., 2019), RepVGG (Ding et al., 2021c), etc.
>
> Ablation study Table in Weakness 2 demonstrates the improvement brought by automatic learning rate tuning (scaling factor) and structural re-parameterization design.
>
> For the training setting of Rep-Adapter, fine-tuning, and linear probing, we follow closely to the original paper of the benchmark VTAB. We use the lightweight sweep setting proposed in VTAB (Zhai et al., 2020) that sweeps the following hyperparameters:
> - Learning rate: {0.1, 0.01}
> - Training schedule: In all cases, we decay the learning rate by a factor of 10 after 1/3 and 2/3 of the training time, and one more time shortly before the end. We try {2500, 5000, 10000} training steps.
>
> Full fine-tuning conducts the same number of epochs as the Rep-Adapter. As we are performing experiments on relatively small datasets with large batch sizes, 10000 training steps are enough when converted to epochs. For example, when performing 1000-shots transfer learning, as we use a total batch size of 256×8=2048, 10000 training steps are equal to 2048×10000/1000=20480 epochs. Therefore, we follow the same setting proposed in VTAB (Zhai et al., 2020).
>
> **Question 2. Ablation of different initialization of $\omega_R$**
>
> Thank you for this suggestion! We perform the experiment by randomly initializing $\omega_R$. As shown in the table, random initializing the tuning branch would clearly damage the performance of Rep_Adapter, demonstrating that the pre-trained initialization is important to the tuning branch. We have added these results and analysis to Appendix A.5 of the rebuttal revision (marked by purple).
>
> | $\omega_R$ init | Caltech101 | CIFAR-100 | DTD | Flowers102 | Pets   |
> | :--- |    :----:   | :----: | :----: | :----: | :----: |
> | random  | 74.8 (5.9)  | 52.5 (1.4) | 53.9 (3.6) |  88.1 (1.1) | 79.1 (4.5) |
> | $\omega_0$ | 84.7 (1.4)  | 59.6 (0.2) | 64.2 (2.4) | 92.7 (0.3) | 90.3 (2.6)|

---

> > ### Comment · Reviewer_xJSa · 2023-11-23
> >
> > Thanks for the detailed response from the authors. My concerns of weakness 3 and weakness 4 have been well addressed. However, I still don't think adding a scaling factor is of significance in the proposed Rep-Adapter since it is a very common technique used in many backbones.  Also, most of the results w/o scaling are very similar to full Rep-Adapter, imho, this makes the statement/section of this work on adaptive learning rate more invalid. Based on these, I will maintain my original rating.

---

### Official Review · Reviewer_XwS8 · 2023-10-31

**Soundness:** 3 good
**Presentation:** 3 good
**Contribution:** 3 good
**Rating:** 8
**Confidence:** 5

**Summary:**

This paper proposes to use Structural Re-parameterization for transfer learning. Specifically, an extra branch comprising a learnable conv layer and a BN is added to the original frozen conv layer during training. After training, such a structure is equivalently transformed into a single conv layer for inference. The effectiveness is explained as adaptively adjusting the equivalent learning rate of filters. Reasonable results are reported.

**Strengths:**

1. The idea of using Structural Re-param for transfer learning is novel.
2. The structural design is easy to understand and thoroughly validated.
3. The results are impressive.
4. The explanation (the structure is equivalent to adjusting lr for different filters) is impressive.

**Weaknesses:**

1. Structural Re-parameterization is not correctly discussed. The authors seem to mistake it with the traditional re-parameterization (e.g., CondConv). Traditional re-parameterization first derives a parameter with some other parameters and uses the derived parameter for computation, for example, a conv layer (y = x conv W) with traditional re-parameterization may compute W = W1 + W2, then y = x conv W. But Structural Re-parameterization uses regular layers during training and converts the structures (i.e., merges some layers) for inference. This work should be categorized into Structural Re-parameterization, but in the paper only "re-parameterization" is used to describe the method. And in Section 2, Structural Re-param should be discussed in a subsection (for example, it should at least mention that Structural Re-param is proposed by [RepVGG] ...) and traditional re-param (e.g., DiracNet, CondConv) should be mentioned in another subsection.

2. The proposition is proved in a vectorized form, which seems a bit messy. I would suggest the authors show a simplified version with a specified arbitrary channel (or a single-channel conv).

I also suggest the authors show the proposition from another equivalent perspective. I guess the authors would like to prove that in the following two simplified scenarios

A. the structure is  y = frozen_conv(x) + trainable_conv(x) * alpha, the learning rate is lamda

is equivalent to

B. the structure is  y = trainable_conv(x), the learning rate is alpha ** 2 * lamda

This may be easier to understand. Then tell the reader what alpha represents (BN.weight / BN.std) so that the reader will understand that the BN realizes adaptive lr. Then naturally discuss the behavior of BN.

**Questions:**

1. How is Rep-Adapter used with transformer? Is it used to replace every linear layer? Is BN still used in this case? BatchNorm-1d or 2d?

2. Is the usage of Rep-Adapter with a linear layer (nn.Linear) simply the same as the usage of a 1x1 conv in a CNN? Their inputs are of different shapes so I wonder if there are some differences.

Please show some code and I will understand it.

---

> ### Author Response · Authors · 2023-11-22
> **Author response to Reviewer XwS8**
>
> Thank you for your recognition of our paper and your detailed and constructive comments. We respond to all the issues you pointed out in detail below. We hope our response and rebuttal revision will address your concerns.
>
> **Weakness 1. Discuss Structural Re-param and Traditional Re-param separately.**
>
> Thank you for this valuable suggestion! We have rewritten this part in Section 2 of the rebuttal revision to discuss Traditional Re-parameterization and Structural Re-parameterization in different subsections. Rep-Adapter belongs to Structural Re-parameterization. Please check the rebuttal revision for the changes (denoted by purple). We hope the revision will address your concern on this.
>
> **Weakness 2. Show a simplified version of the proposition with a single channel. Also, show the proposition in a simplified scenario for better understanding.**
>
> Thank you for this suggestion! Following your suggestion, we present a simplified case of Proposition 3.1 and write a proof with single channel. The Proposition and proof are added to Appendix A.1 (marked as purple) of the rebuttal revision to provide a better understanding for the readers.
>
> **Question 1. Details of the Rep-Adapter used with transformer.**
>
> Thank you for this question. We replace every linear layer in transformers with Rep-Adapter, including the ones in multi-head attention and MLP. In transformers, we use Rep-Adapter *without* BN. The scaling factors in transformers are implemented separately as learnable parameters, instead of directly using the weight of BN as in ConvNets (code is shown in Question 2). The original LayerNorm layers in the transformer are not part of Rep-Adapter and are left intact.
>
> **Question 2. Details of the Rep-Adapter used with linear layer. (code)**
>
> Thank you. The usage of Rep-Adapter with a linear layer (`nn.Linear`) is very similar to the usage of Rep-Adapter with a 1x1 conv in a CNN. The implementation is changed accordingly as their inputs are of different shapes.
>
> Here, we show an example of the implementation of RepAdapter in PyTorch as you suggested:
>
> ```
> class RepAdapter(nn.Module):
>     def __init__(self, in_dim, out_dim, init_values):
>         super().__init__()
>         self.branch_1 = nn.Linear(in_dim, out_dim)    # this branch is frozen
>         self.branch_2 = nn.Linear(in_dim, out_dim)
>         self.gamma_1 = nn.Parameter(init_values * torch.ones((out_dim)),requires_grad=True)
>         self.gamma_2 = nn.Parameter(init_values * torch.ones((out_dim)),requires_grad=True)
>     def forward(self, x):
>         # x shape is : batch, length, dim
>         x1 = self.gamma_1 * self.branch_1(x)
>         x2 = self.gamma_2 * self.branch_2(x)
>         return x1 + x2
> ```
> In transformer, `nn.Linear(in_dim, out_dim)` is replaced by `RepAdapter(in_dim, out_dim)`.

---

### Official Review · Reviewer_ZcYB · 2023-10-31

**Soundness:** 3 good
**Presentation:** 3 good
**Contribution:** 3 good
**Rating:** 6
**Confidence:** 4

**Summary:**

The paper addresses the challenges in transfer learning, emphasizing the impact of factors like dataset size and label fraction on different transfer learning protocols. It highlights the efficacy of linear probing and fine-tuning in semi-supervised and fully-supervised scenarios, respectively. The proposed solution, Rep-Adapter, introduces an approach by adding a learnable side branch alongside a frozen pre-trained branch. This strategy aims to strike a harmonious balance between pre-trained and fine-tuned weights. To simplify the process, learnable hyper-parameters for each layer are introduced, eliminating the need for manual tuning. Additionally, a re-parameterization method is employed during inference to merge the two branches while preserving the structure of the pre-trained model.

**Strengths:**

* The combination of learnable and frozen branches to find the balance between pre-trained weights and fine-tuned weights.
* Learnable hyper-parameters for each layer, reducing the need for manual adjustments.
* Efficient re-parameterization during inference, ensuring minimal additional computational cost.

**Weaknesses:**

* Increased computational cost during training due to the addition of learnable branches and hyper-parameters, especially for heavy-weight models.
* In theory, the final results of the proposed fintuning can be achieved by the traditional fintuning, i.e. the difference of original weights and the final weights can be achieved by traditional fintuning. Thus, it is arguable this method is significantly different or better than traditional finetuning.

**Questions:**

Could it be applied on other architectures, like attention-based ones?

---

> ### Author Response · Authors · 2023-11-22
> **Author response to Reviewer ZcYB**
>
> Thanks for your detailed and constructive comments and your recognition on our paper. We respond to all the issues you pointed out in detail below. We hope our response and rebuttal revision will address your concerns.
>
> **Weakness 1. Increased training cost.**
>
> Thank you for pointing that out! We would like to address this concern from two aspects.
>
> 1. As we do not increase the depth of the network or the total training steps during training, the computational overhead on modern GPUs is much lower than expected. We have tested our training cost on A100 GPUs. The training cost of Rep-Adapter is around 1.2× compared to fine-tuning. In contrast, other adaptive lr method, e.g. AutoLR (Ro & Choi, 2021), requires more than 2× training cost for repeated training.
>
> 2. We would like to point out that our method does not aim at fast adaptation, but aims at the transfer learning performance, which is in line with L2-SP, DELTA, AutoLR, and SpotTune. Also, previous structural re-parameterization works usually achieve performance improvements at the cost of training efficiency. Among them, Rep-Adapter achieves remarkable performance with a relatively small increase in training costs. For example, Rep-Adapter cost only around 1.2× during training, bringing 11.1% mean top-1 accuracy improvements on supervised VTAB-1k compared to Visual Prompt Tuning (Table 4 in the paper).
>
> **Weakness 2. Difference and advantage over fine-tuning.**
>
> Thank you for this valuable comment! We would like to point out that the final results of our tuning method could not be easily achieved by traditional fine-tuning. The reasons are three-fold.
>
> 1. During Rep-Adapter tuning, every filter of ConvNets can be seen as tuning at a different learning rate, while in traditional fine-tuning, every filter uses the same predefined learning rate.
>
> 2. Compared to fine-tuning with filter-wise learning rate settings, our Rep-Adapter is an automatic transfer learning protocol that saves the tediously searching for hyper-parameter settings.
>
> 3. Rep-Adapter also optimizes the pre-trained model in a larger network capacity and complexity compared to fine-tuning, which is an advantage of structural re-parameterization. This is similar to the advantage of structural re-parameterization training over traditional training and has been generally proved by the success of ACNet (Ding et al., 2019), RepVGG (Ding et al., 2021c), etc. This advantage of structural re-parameterization can also be proved by our ablation study (c), where we study the important component of structural re-parameterization (Ding et al., 2021c;b), the batch-wise statistics in BN during training.
>
> Given the above reasons, we believe our Rep-Adapter has a clear difference and advantage over fine-tuning.
>
> **Question 1. Application on other architectures.**
>
> Thank you! Although Rep-Adapter is mainly designed for ConvNets, it can also be applied to attention-based architectures. For example, in Section 4.5, we explore our Rep-Adapter on CLIP with a vision transformer as the visual encoder. We apply Rep-Adapter on every fully-connected layer in the visual encoder of CLIP-ViT-B/32. Our method outperforms CoOp by 1.6% on average.

---

### Author Response · Authors · 2023-11-23
**General Response to All Reviewers**

We thank all the reviewers for their time, insightful suggestions, and valuable comments. We are grateful for the positive recognition of the reviewers on our technical contribution (ZcYB and XwS8), novelty (XwS8), theoretical explanation (XwS8), extensive experiments (XwS8 and xJSa), and good results (XwS8). We respond to each reviewer's comments in detail below. We have also revised the main paper and the supplementary material according to the reviewers' suggestions. The main changes are listed as follows:

- We have rewritten the literature review of re-parameterization in Section 2 to discuss Traditional Re-parameterization and Structural Re-parameterization in different subsections as kindly suggested by reviewer XwS8.

- We have presented a simplified version of Proposition 3.1 and wrote a proof with a single channel as suggested by reviewer XwS8. The Proposition and proof are added to Appendix A.1 of the rebuttal revision to provide a better understanding for the readers.

- We have added an additional ablation study on the scaling factor to study the relation between the scaling factor and Rep-Adapter. The results and analysis are added to Appendix A.5 to study the valuable question raised by reviewer xJSa. The result shows that the scaling factor is an important component of Rep-Adapter. However, the results also show that it needs the other design of Rep-Adapter to be effective.

- We have also added an additional ablation study on randomly initializing the weight of the tuning branch of Rep-Adapter as suggested by reviewer xJSa. The results and analysis are added to Appendix A.5. The results show that randomly initializing the tuning branch would clearly damage the performance of Rep_Adapter, demonstrating that the pre-trained initialization is important to the tuning branch.

All of the changes we made in the rebuttal revision are marked in purple. We hope our response and rebuttal revision will address the reviewers' concerns. We look forward to further discussion with all the reviewers.

---

### Meta-Review · Area_Chair_DAST · 2023-12-18

**Metareview:**

This paper proposes a fine-tuning procedure utilizing an extra branch with convolution and BN layers (added to the frozen pre-trained network) that is then converted into a single conv layer using structural re-parameterization. Analysis is shown connecting the method to adaptive learning rates without the need to decide them via hyper-parameter tuning. Results on many/low/few-shot finetuning are shown across a range of datasets, with some improvement.

  The paper received mixed reviews, with the authors appreciating the use of structural re-parameterization for this task, but a number of weaknesses were identified including 1) Increase in computation cost during training, 2) Writing and clear connections to the different types of related work (structural vs. traditional re-parameterization), 3) The novelty compared to prior works such as Rebuffi et al., and 4) Significance of the results. The authors provided a rebuttal, including run-time comparison (with the method taking 1.2x that of traditional fine-tuning), new ablations, and a clearer positioning.

  After rebuttal, the scores remain mixed at 6, 8, and 3, with the reviewer xJSa still expressing concerns about the significance of the scaling factor and notably reviewer ZcYB (who gave it a six) also expressing the concern of significance of improvements given the costs. After looking at the paper, reviews, rebuttals, and discussion, I agree with these that overall the paper does not warrant acceptance due to these. Specifically, the method itself is relatively incremental compared to the prior works mentioned above (with only some technical details being different, e.g. addition of non-linearity) and the overall performance (as described in Table 1) are also incremental in many cases with only small mean differences given the additional overhead and complexity. In summary, the various small incremental advancements do not result in a substantial contribution that can be accepted. We encourage the authors to strengthen these aspects for a future submission.

**Justification For Why Not Higher Score:**

Overall, even the positive reviewer (rating it a 6) mentioned that the contributions are not significant and both the conceptual and empirical contributions are very incremental.

**Justification For Why Not Lower Score:**

N/A

---

### Decision · Program_Chairs · 2024-01-16

Reject